# A Comparative Study of RoPE-based Positional Encodings from A Scaling Perspective

**Zhu Zhang**
Tsinghua University

**Tianxing Yang**
Tsinghua University

**Zihan Yan**
Tsinghua University

## 1 Background and Motivation

Transformers [11], as the backbone of Large Language Models (LLMs), have become the dominant architecture for natural language processing tasks. However, their quadratic computational complexity makes training on long sequences inefficient and resource-intensive. A common solution involves pre-training the model on shorter sequences (e.g., 4k tokens) to develop initial capabilities, followed by further pre-training on longer sequences (e.g., 32k tokens) to extend the context window. This approach is feasible since the additional pre-training requires significantly fewer tokens than the initial phase, enabling effective length extrapolation.

Positional encoding is crucial for length extrapolation. Rotary Positional Encoding (RoPE) [8] has become popular due to its superior performance. Typically, RoPE's base frequency is set to 10,000, and models are pre-trained on sequences of 4k tokens. However, an out-of-distribution (OOD) issue can arise when input lengths exceed the original context window without additional measures. To address this, several RoPE variants have emerged, such as PI [2], ABF, NTK [3], and YaRN [5]. Despite differing in form, these variants share the goal of introducing a scaling mechanism to improve performance on extended contexts.

Despite these efforts, it remains unclear which variant is consistently superior or why RoPE is effective in Transformer models. Therefore, our work aims to address the following research questions:

- In long-context scenarios, how do different RoPE-based positional encoding schemes compare in terms of principles and performance?
- What underlying mechanism makes RoPE effective in LLMs?

Ultimately, our goal is to propose a new positional encoding that outperforms existing methods in long-context scenarios.

## 2 Preliminary

In this section, we present the formal definitions for Attention [11] and Rotary Positional Encoding (RoPE) [8].

Attention operates over a sequence of $C$ embeddings, represented as $\mathbf{X} = [\mathbf{x}_1, \mathbf{x}_2, \ldots, \mathbf{x}_C]^\top \in \mathbb{R}^{C \times d}$, where $d$ is the model dimension. Learned weight matrices $\mathbf{W}_v \in \mathbb{R}^{d \times d_k}$, $\mathbf{W}_q \in \mathbb{R}^{d \times d_k}$, and $\mathbf{W}_k \in \mathbb{R}^{d \times d_k}$ are applied to these inputs, where $d_k$ is the dimension of the projected embeddings. The attention mechanism computes the attention matrix and uses it to produce a weighted sum of the value vectors, as follows:

$$\text{Attention}(\mathbf{Q}, \mathbf{K}, \mathbf{V}) = \text{softmax}\left(\frac{\mathbf{Q}\mathbf{K}^\top}{\sqrt{d_k}}\right)\mathbf{V}.$$

Submitted to AML course at Tsinghua University, 2024 Fall.

In basic attention, the query, key, and value matrices are computed as $\mathbf{Q} = \mathbf{XW}_q$, $\mathbf{K} = \mathbf{XW}_k$, and $\mathbf{V} = \mathbf{XW}_v$. However, this formulation does not explicitly account for the relative positions of keys and values.

RoPE [8] addresses this by encoding positional information through a phase rotation applied to each element of the embedding vectors. Formally, we define a transformation $\mathbf{f}$ as:

$$\mathbf{f}_{\mathbf{W}}(\mathbf{x}_i, \boldsymbol{\theta}) = \mathbf{R}(\boldsymbol{\theta}, i)\mathbf{W}^\top \mathbf{x}_i,$$

where $\mathbf{x}_i \in \mathbb{R}^{d_k}$ is the embedding at position $i$, $\mathbf{W}$ is a projection matrix, and $\boldsymbol{\theta} \in \mathbb{R}^{d_k/2}$ is a frequency basis. The rotary transformation matrix $\mathbf{R}(\boldsymbol{\theta}, i)$ is defined as:

$$\mathbf{R}(\boldsymbol{\theta}, i) = \begin{pmatrix} \cos i\theta_1 & -\sin i\theta_1 & \cdots & 0 & 0 \\ \sin i\theta_1 & \cos i\theta_1 & \cdots & 0 & 0 \\ \vdots & \vdots & \ddots & \vdots & \vdots \\ 0 & 0 & \cdots & \cos i\theta_{\frac{d_k}{2}} & -\sin i\theta_{\frac{d_k}{2}} \\ 0 & 0 & \cdots & \sin i\theta_{\frac{d_k}{2}} & \cos i\theta_{\frac{d_k}{2}} \end{pmatrix}.$$

This matrix has the property that $\mathbf{R}(\boldsymbol{\theta}, n - m) = \mathbf{R}(\boldsymbol{\theta}, m)^\top \mathbf{R}(\boldsymbol{\theta}, n)$, based on Ptolemy's identity. As a result, the query-key product between two positions $m$ and $n$ is redefined as:

$$\mathbf{q}_m^\top \mathbf{k}_n = \mathbf{f}_{\mathbf{W}_q}(\mathbf{x}_m, \boldsymbol{\theta})^\top \mathbf{f}_{\mathbf{W}_k}(\mathbf{x}_n, \boldsymbol{\theta}),$$

where the relative positional information $n - m$ is implicitly encoded in the attention score through the query-key interaction.

In the standard RoPE transformation, the components of $\boldsymbol{\theta}$ are defined as $\theta_j = b^{-\frac{2j}{d_k}}$ with a base frequency of $b = 10,000$.

## 3 Related Work

**Positional Encoding**. Positional encoding is a crucial component in the transformer architecture [12]. RoPE, introduced by [9], applies sinusoidal rotations to hidden representations before the self-attention mechanism. Numerous RoPE-based improvements have since been developed. ABF [13, 6] involves increasing the base frequency $\theta$ in RoPE (e.g., from 10,000 to 500,000), as seen in recent LLaMA 3 models [10]. This adjustment reduces the attention decay effect, enabling the model to handle longer contexts. Other works have highlighted that RoPE's performance degrades in out-of-distribution (OOD) settings. For example, NoPE [4] introduces a causal mechanism to learn positional encodings. Additionally, [7] proposed randomized positional encodings, claiming that they improve OOD performance and enable the model to capture relationships over longer text spans. Recent work [1] provides a comprehensive analysis of existing positional encoding methods and offers explanations for their performance.

## 4 Initial Methodology

We intend to undertake the following steps:

- **Data recipe and upsampling**: We will systematically design the data recipe and determine an optimal mix of short and long data sequences. This approach ensures that the continuous pre-training phase utilizes a scientifically balanced mixture of data from different domains, facilitating effective long-context extrapolation experiments.
- **Scaling experiments**: We will conduct scaling experiments on models with varying parameter sizes to assess the performance of different positional encoding methods. These experiments will provide a solid foundation for the development of our own positional encoding approach.

Through these steps, we aim to identify the key factors influencing the efficacy of different positional encoding schemes and establish a robust experimental basis for proposing a more effective positional encoding method.

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
