# OpenReview forum: "【Proposal】A Comparative Study of RoPE-based Positional Encodings from A Scaling Perspective"
_tsinghua.edu.cn/THU/2024/Fall/AML — THU 2024 Fall AML Submission_

### Official Review · ~Thomas_Adler2 · 2024-11-06
**Clear explanation**

**Rating:** 10
**Confidence:** 4

**Review:**

Overall the proposal is clear and well researched, it is clear what the problem is, why it is important what it will attempt to do. A point of improvement would be a more specific methodology section to better understand the specifics.

---

### Official Review · ~Un_Lok_Chen1 · 2024-11-08
**A Comparative Analysis of RoPE Variants**

**Rating:** 8
**Confidence:** 4

**Review:**

Summary:

This project aims at comparing the different Rotary Positional Encoding (RoPE) variants by conducting a relatively large scale of experiments and investigating the factors behind the efficacy of certain methods, which may aid in their ultimate goal to propose a new positional encoding strategy. The authors also attempt to design a dataset recipe to provide a experimental basis with mixed-length language sequences.

Pros:

1. The research objective is well stated and clear to understand. The research problem is also considered as meaningful, with the potential to provide more insights on how positional encoding method will affect model performance.
2. Although not necessary according to the proposal guideline, the authors introduce the preliminaries on the attention mechanism and the RoPE method with essential formulas.

Cons:

A. Major issues:

1) In the Background and Motivation section, the authors first introduce how the two-stage pretraining scheme can enable length extrapolation while reducing training cost. This part seems irrelevant to the rest of the paper that discusses about positional encoding.

2) The comparison between different RoPE variants may require a large amount of LLM training, are the resource requirement and workload feasible for a course project?

3) Besides an experimental approach, if the authors aim at explaining the efficacy of the current RoPE method and developing a new positional encoding method, are there any theoretical foundations that may lay the ground for devising the new technique?

B. Minor issues:

1) In the second paragraph of the Background and Motivation section, the authors mention that variants of RoPE introduce a “scaling mechanism” to improve performance. It is better to elaborate more on what the “scaling mechanism” concept means.

2) Consider providing more details on how the experiments will be conducted (e.g. using what datasets, using what model architecture, evaluating on what types of task, evaluate on what specific aspects of the method performance, using what evaluation metrics).

---

### Official Review · ~Guangjie_Xu1 · 2024-11-09

**Rating:** 9
**Confidence:** 5

**Review:**

This proposal provides a compelling overview of the challenges in scaling RoPE-based positional encodings for long-context tasks within transformer models. By addressing the limitations in quadratic complexity and the out-of-distribution issues when extending context windows, the authors clearly identify the research gap. Their systematic approach to comparing RoPE variants and investigating their underlying mechanisms is well-structured. The focus on both practical scaling experiments and theoretical exploration of RoPE’s efficacy adds depth to the study. Overall, this work promises valuable insights for advancing transformer efficiency in handling extended contexts and could potentially shape future positional encoding strategies.

---

### Official Review · ~Lu_Fan_DB1 · 2024-11-09
**Review of the Proposal: "A Comparative Study of RoPE-based Positional Encodings from A Scaling Perspective"**

**Rating:** 7
**Confidence:** 3

**Review:**

Strengths:
Relevance and Novelty: The proposal addresses a timely and relevant issue in the field of natural language processing, specifically within transformer architectures. The focus on comparing RoPE-based positional encoding schemes for long-context language models is well-motivated, especially given the practical importance of improving efficiency in long-context processing.
Clear Research Questions: The authors have defined clear research questions regarding the performance and mechanisms of different RoPE-based positional encodings in long-context scenarios. These questions provide a focused scope and contribute to the clarity of the proposed study.
Methodological Approach: The planned methodological approach, including scaling experiments and data recipes, reflects a structured and systematic strategy for addressing the research questions. The proposal also outlines a balanced mixture of short and long sequence data, suggesting a thoughtful design for testing length extrapolation.
Areas for Improvement:
Depth in Preliminary Analysis: While the proposal covers background information on RoPE and transformer mechanisms, further elaboration on the specific limitations of existing RoPE-based encoding methods would strengthen the reader's understanding of the need for this comparative study. This could include more concrete examples of how RoPE variants handle different long-context scenarios.
Related Work Discussion: Although the proposal references relevant prior works, including PI, ABF, and YaRN, a more detailed comparative analysis of these methods and their limitations could enrich the review of related work. Explicitly contrasting their performance in long-context settings would better justify the choice of methods for comparison.
Experimental Design: While the proposal briefly outlines scaling experiments, additional details on specific metrics (e.g., accuracy, computational efficiency) and benchmarks would provide a clearer picture of the study’s evaluation criteria. Including potential baseline models or configurations for each RoPE variant would further enhance the robustness of the methodology.
Additional Suggestions:
Mechanistic Insight: To strengthen the theoretical contribution, the proposal could aim to not only compare empirical performance but also to explore the underlying mechanisms of RoPE variants in more detail. A discussion on potential hypotheses regarding why certain variants might perform better in long contexts would add depth to the analysis.
Limitations and Challenges: Including a brief acknowledgment of the limitations or challenges anticipated in the comparative analysis, such as computational resource requirements or difficulties in replicating previous results, would provide a more comprehensive view of the study’s scope.
Overall Assessment:
This proposal presents a relevant and well-motivated study in the field of positional encoding for long-context transformers. By addressing the suggested improvements, especially in detailing related works and providing a more rigorous experimental framework, this study has the potential to contribute valuable insights into RoPE-based positional encodings for language models.

---

### Official Review · ~Diego_Cerretti1 · 2024-11-10
**Clear and insightful**

**Rating:** 9
**Confidence:** 4

**Review:**

The authors propose a comparison between RoPE-based positional encodings in long-context scenarios, aiming to identify the properties that improve the transformers' performance on extended sequences. The proposal is backed by clear motivations. The study is highly relevant and the methodology is well-structured. I would suggest adding further details on the evaluation metrics for the experiments.

---

### Official Review · ~Zhijie_shen3 · 2024-11-10
**Peer Review**

**Rating:** 9
**Confidence:** 4

**Review:**

### Summary
This proposal is well-structured, with a clear motivation and research plan. The study addresses an important issue in scaling Transformers for long-sequence tasks.

### Comments
1. The motivation is well-articulated, highlighting the limitations of Transformers in handling long sequences and the potential of RoPE-based encoding.
   - **Suggestion**: Provide more details on the specific advantages of RoPE over existing methods to better emphasize the need for this study.

2. The research plan is well-organized, with clear steps for experimentation. The use of mixed-length datasets to test scalability is a strong point.
   - **Suggestion**: Include more details about the specific models, parameters, and evaluation metrics that will be used. This would improve the transparency.

---

### Official Review · ~Chengming_Shi1 · 2024-11-11

**Rating:** 7
**Confidence:** 4

**Review:**

### Summary

The proposal “A Comparative Study of RoPE-based Positional Encodings from A Scaling Perspective” aims to investigate the effectiveness of various Rotary Positional Encoding (RoPE) methods in the context of Large Language Models (LLMs) dealing with long sequences. The study plans to compare different RoPE variants, such as PI, ABF, NTK, and YaRN, to determine their performance in long-context scenarios and to understand the underlying mechanisms that make RoPE effective. The research also seeks to propose a new positional encoding method that surpasses existing techniques for extended contexts.

### Pros

1. **Addressing a Key Challenge**: The proposal tackles the significant issue of handling long sequences in Transformers, which is critical for the advancement of LLMs.
2. **Comparative Analysis**: By comparing different RoPE variants, the study aims to provide a comprehensive understanding of their relative strengths and weaknesses.
3. **Focus on Scaling**: The emphasis on scaling experiments will offer insights into how positional encoding methods can be adapted to models of varying sizes.
4. **Methodological Rigor**: The methodology, including data recipe design and upsampling, is well thought out and aims to ensure the robustness of the experiments.
5. **Potential for Innovation**: The goal of proposing a new positional encoding method could lead to significant improvements in the field of natural language processing.

### Cons

1. **Complexity of Experiments**: Conducting a comparative study across multiple RoPE variants and various model sizes may be resource-intensive and complex.
2. **Potential Overfitting**: The study may risk overfitting to specific experimental conditions without generalizing well to real-world applications.
3. **Dependence on Pre-training**: The reliance on pre-training for effective length extrapolation may limit the immediate applicability of the findings to scenarios where extensive pre-training is not feasible.
4. **Out-of-Distribution Concerns**: The proposal acknowledges OOD issues but does not detail how these will be mitigated in the study.
5. **Undefined Success Metrics**: The proposal does not clearly define the metrics for evaluating the performance of different positional encoding schemes.

---

### Official Review · ~Mingdao_Liu1 · 2024-11-11
**Review for "A Comparative Study of RoPE-based Positional Encodings from A Scaling Perspective"**

**Rating:** 8
**Confidence:** 3

**Review:**

This proposal aims to compare different RoPE schemes on performance and principle and explore the reason why RoPE is effective for LLMs. The proposal plans to (1) design a data recipe and determine an optimal mix of short and long data sequences in the continuous pre-training stage and (2) test the scaling property of different RoPE methods on models of different sizes.

Pros:
1. The proposal is well-written and includes all required sections
2. The research problems are important in long-context LLMs, and the results may be insightful for future research in positional encoding.

Cons:
1. The proposed plan includes continuous pre-training experiments, which typically require considerable computation resources.

---

### Official Review · ~Wuqian1 · 2024-11-11
**The proposal presents a well-structured and detailed study on the effectiveness of Rotary Positional Encoding (RoPE) in Large Language Models (LLMs), focusing on scaling perspectives. The quality of the work is evident in its thorough background, clear problem statement, and the methodology outlined for the comparative study.**

**Rating:** 9
**Confidence:** 4

**Review:**

Pros:
    1.Comprehensive Background:The proposal provides a thorough background on Transformers and the importance of positional encoding.
    2.Clear Research Questions: The research questions are well-defined and directly address the need for improved positional encoding in LLMs.

Cons
    1.Lack of Preliminary Results: The proposal does not include any preliminary results, which could have provided an initial validation of the research questions.
    2.Limited Discussion on Challenges: The potential challenges in implementing the proposed methodology are not discussed in detail.

---

### Official Review · ~KAI_JUN_TEH1 · 2024-11-11
**Innovative ideas & clear research directions**

**Rating:** 9
**Confidence:** 4

**Review:**

This paper presents a comparative study of different Rotary Positional Encoding (RoPE)-based positional encoding schemes from a scaling perspective. First, the paper compares different RoPE-based positional encoding schemes in long-context scenarios to understand their principles and performance. Based on the observed phenomena and results, propose a better method. However, could I learn more about how to systematically design a data recipe in experiments?

---

### Official Review · ~Zheng_Jiang2 · 2024-11-11
**Novel and meaningful proposal**

**Rating:** 9
**Confidence:** 4

**Review:**

**Summary:**
This paper presents a comparative study of Rotary Positional Encoding (RoPE) variants within the context of scaling transformer models for long sequence processing. The authors aim to identify why RoPE is effective in LLMs and thus propose a new positional encoding method that outperforms existing methods in handling extended contexts.

**Strengths:**
1. The paper addresses a significant challenge in the field of natural language processing, which is the inefficiency of transformers with long sequences. The study of RoPE and its variants is timely and relevant to the current research landscape.
2. The paper provides a thorough analysis of different RoPE-based positional encoding schemes, which is crucial for understanding their comparative performance and underlying mechanisms.
3. The proposed methodology for data recipe and upscaling experiments seems systematic and well-designed to facilitate effective long-context extrapolation experiments.

**Weaknesses:**
1. The paper does not provide enough experimental details. I wonder which model architecture will you choose and whether the final methods can be generalized to different types of transformer models or if they are specific to certain architectures.
2. The authors plan to conduct scaling experiments on models with varying parameter sizes, which will consume considerable resources. Is it possible to just validate on small models and then provide insights for the design of large models?